# Study on the Impact Damage Characteristics of Transplanting Seedlings Based on Pressure Distribution Measurement System

Hongbin Bai [ID], Xuying Li *, Fandi Zeng [ID], Ji Cui and Yongzhi Zhang

College of Mechanical and Electrical Engineering, Inner Mongolia Agricultural University, Hohhot 010018, China
* Correspondence: lixuy2000@imau.edu.cn

**Abstract:** Collision is one of the main causes of mechanical damage to the seedling during transplanting. To reveal the impact damage characteristics of plug seedlings, the kinetics equations of seedling collision were established based on Hertz's contact theory, and the kinematic characteristics, elasto-plastic deformation, and collision damage during seedling collision were analyzed using high-speed photography. Using the Tekscan pressure distribution measurement system, the significant levels of various factors that affect impact peak force (IPF) and damage of seedling pot (DSP) were studied, the change rule of contact pressure distribution of seedlings under significant factors was measured, and a regression model between IPF and DSP was established. The results showed that collision material, drop height and seedling pot size had significant effects on IPF and DSP. The contact pressure area of different collision materials from large to small was soil block, steel, and ABS plastic. The contact pressure area of seedlings of different pot sizes was big, medium, and small in descending order. At a dropping height of 50~350 mm, a contact pressure > 10 kPa accounted for the major contact pressure area, which is the main reason for collision damage of the seedling pot. Linear regression models between IPF and DSP under different factors were established, and the determination coefficients ($R^2$) were 0.98 and 0.94, respectively. The results provided a theoretical basis for understanding the collision damage mechanism of the plug seedling and how to reduce damage during transplanting.

**Keywords:** transplanting; seedling; impact; damage; contact pressure; force

## 1. Introduction

Transplanting is an agriculture task that involves transferring and planting seedlings that are grown uniformly in a nursery to a field [1]. The use of seedling transplanting technology for crop production can increase the land multiple cropping index, improve crop yield and quality, and mechanized transplanting can further improve planting quality and reduce production costs [2,3]. Seedlings grow on a substrate and absorb nutrients from the substrate. Seedling roots are mixed with the substrate to form a pot with special-strength elasticity for transplanting in the field [4,5]. During mechanized transplanting, the seedling needs to go through the processes of picking up, feeding, carrying, and planting [6]. The collision, impact and squeeze between the seedling and the key components of the machines are unavoidable, which can easily cause mechanical damage to the seedling pot. Mechanical damage will seriously affect the survival and quality of the seedlings and cause economic losses [7,8]. Therefore, there is an urgent need to study the mechanical characteristics of plug seedlings.

At present, the research on the pot damage and mechanical properties of plug seedlings mainly focuses on the seedling-picking process [9–11]. For simulation, Wang et al. [12] used finite element analysis software to study the mechanical effects of three end-effector structures on the seedling pot body from a microscopic scale. Gao et al. [13] conducted a force analysis of the plug seedling lift-off process based on the discrete element method. However, in the above studies, the contact conditions between the seedling pickup mechanism and the seedling pot were relatively fixed, and the influence of the contact conditions

on the fragmentation mechanism and strength of the seedling substrate particles was not taken into account [14]. Tian et al. [15] proposed a new multi-needle seedling gripper, and explored the influence of different needle diameters, insertion depths, and insertion and grabbing speeds on the substrate integrity through the response surface method. Jiang et al. [16] studied the effects of the substrate ratio, moisture content of the seedling pot, and seedling mass on the seedling extraction rate and the seedling pot breakage rate. Mohamed et al. [17] investigated the effects of different soil moisture content and pickup speed on pickup force and lump damage during the transplanting of seedlings. However, studies on the mechanical characteristics of the drop impact of plug seedlings during feeding, carrying, and planting are relatively rare. Liu et al. [18] analyzed the influence of seedling drop height and impact angle on impact force through a contact mechanics model for the collision between seedlings and transplanters. Xiang et al. [19] established a finite element model of the collision between plug seedlings and planters using ANSYS software and found that pressure distribution is an important foundation for judging the yield states of seedlings. Wang et al. [6] and Liu et al. [20] analyzed the collision motion between seedlings and transplanters and concluded that when the relative motion speed of seedlings is too large, the impact peak force will exceed the yield strength of the seedling pot, resulting in the deformation and damage of the seedling pot. These studies on seedling collision are based only on theoretical analysis and simulation, which are carried out under ideal conditions. Nevertheless, to the best of the author's knowledge, there are no publications of studies on the collision damage characteristics and contact pressure distribution of plug seedlings available. The contact pressure distribution of hole tray seedlings under load is directly related to the pot damage of seedlings during transplanting [21]. At present, there are several methods commonly used to study the impact damage characteristics of agricultural and biological objects, namely, methods of acceleration sensing [22,23], ultrasonic technology [24], the pressure-sensitive film technique [25,26], and the Tekscan pressure distribution measurement system [27,28]. The acceleration sensing method is an indirect measurement, and its data processing errors are high. The method of ultrasonic technology has acceptable sampling accuracy, but it is impossible to observe the collision process intuitively. Both pressure-sensitive technology and the Tekscan pressure distribution measurement system are nondestructive mechanical measurement methods that effectively avoid the problem of insufficient sensitivity, and the pressure distribution between various contact surfaces can be visually measured and analyzed under static and dynamic load conditions. DeMarco's research concluded that the Tekscan pressure measurement system could be used to yield results that are equally or more accurate than those obtained using the pressure-sensitive film technique [29]. Therefore, the Tekscan pressure distribution measurement system was used to analyze the contact pressure distribution of plug seedlings during the collision process and explore the impact damage mechanism of seedlings.

This paper used T562 oil sunflower plug seedlings as the research object, The kinetics equations of the seedling collision were established based on Hertz's contact theory, and the impact test platform of plug seedlings was built using high-speed photography and the Tekscan pressure distribution measurement system. The kinematical characteristics, elastoplastic deformation and pot damage were analyzed, the significant level of various factors that affected DSP and IPF were studied, and a regression model between IPF and DSP was established. The study results can provide a basis for exploring the impact damage mechanisms of plug seedlings and reducing the pot damage of seedlings in a collision.

## 2. Materials and Methods

### 2.1. Test Material

A typical variety of oil sunflower seedlings, Tonghui 562, was selected as the research subject, as shown in Figure 1. The seedlings were incubated in a 50-, 72-, and 105-pore seedling plastic tray, and the age of the seedlings was 25 days. The moisture content of the seedling substrate was measured using the drying method [30], and the range of

variation was obtained at between 58% and 63%. The seedling tray length and width was 540 × 280 mm. Each pore of the seedling tray was shaped like an inverse quadrangular platform. The seedling substrate was a mixture of charcoal, perlite, and vermiculite. The volume ratio was as follows: 3:1:1 of charcoal, perlite, and vermiculite, respectively.

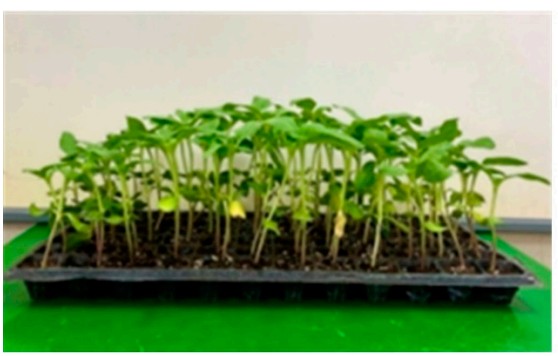 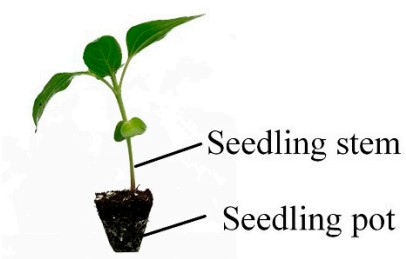

**Figure 1.** Oil sunflower seedlings.

*2.2. Seedling Impact Test System*

The seedling impact test system is shown in Figure 2. The testing instruments included a bracket, the collision material, a height ruler, a gripper, a data acquisition device (EH-2, Tekscan, Inc., Boston, MA USA), a high-speed digital camera, a computer, and an ultra-thin tactile pressure sensor (Tekscan 5250). The high-speed digital camera was fixed on the camera stand for recording with a total of 1279 frames per second and an exposure time of 4.5 ms. According to the actual mechanical transplanting of the seedling collision objects in the field, the following test collision materials were selected: steel, ABS plastic, and soil blocks. The material properties are shown in Table 1. The type of soil block used was sandy loam, and its characteristic parameters were: a moisture content of 14.52% and firmness of 90–100 N/cm$^2$. During the test, the sensor was placed on the top surface of the collision materials. The seedling was placed at a certain height and released to collide with the collision material. The test system could detect the change in collision load during the collision process. At the same time, the high-speed digital camera was used to capture the kinematic characteristics of the seedlings.

The seedling pot is a mixture of roots and substrate, in which the substrate mainly wraps the roots to prevent root leakage or damage. Compared with the recovery time required by normal seedlings after transplanting, the recovery time required by seedlings with damaged roots is usually 3~5 days longer, and the recovery time required for seedlings with root leakage is usually 1~3 days longer. Therefore, the root system of seedlings should be wrapped and moved by a complete substrate during transplanting, and the collision damage of the seedling pot should be as small as possible [31]. In this study, the mass of seedlings before and after dropping was recorded, and the damage to the seedling pot (DSP) was calculated. Higher values of DSP indicate worse substrate encapsulation, that is, more probability of seedling root damage. Equation (1) is the formula used for calculating the DSP.

$$DSP = \left(1 - \frac{M_2}{M_1}\right) \times 100\% \tag{1}$$

where $M_1$ and $M_2$ are the mass of the seedlings before and after the collision, respectively.

**Table 1.** Properties of collision material.

| Materials | Density/(g/cm$^3$) | Elastic Modulus/Gpa | Poisson's Ratio |
|---|---|---|---|
| Steel | 7.85 | 182 | 0.3 |
| ABS plastic | 1.07 | 2.2 | 0.39 |
| Soil blocks | 1.45 | 0.00284 | 0.42 |

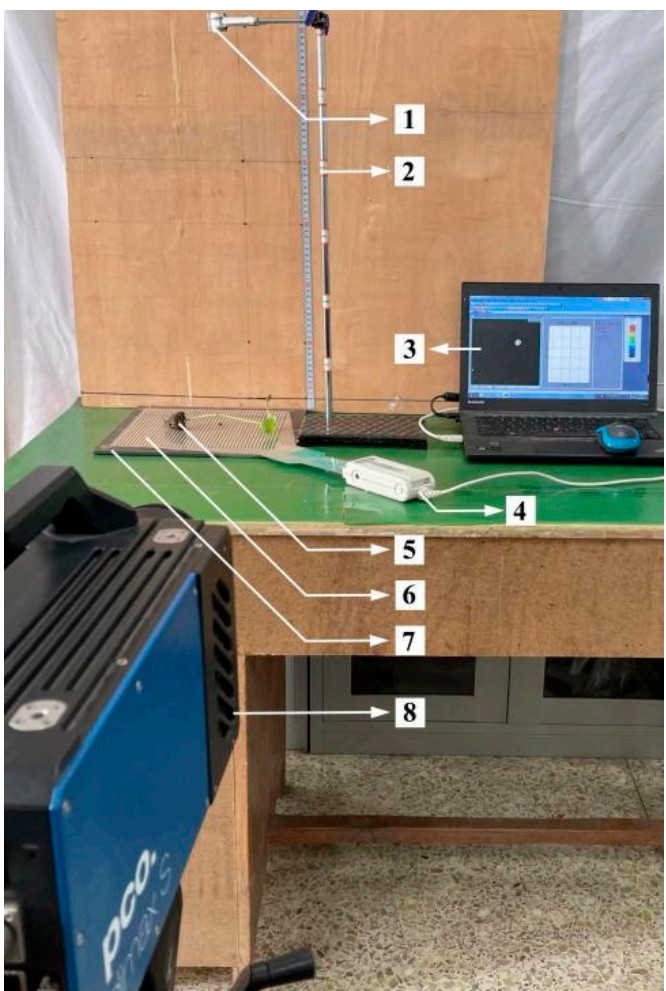

**Figure 2.** Seedling impact test system: 1. gripper; 2. bracket; 3. computer; 4. data acquisition and device; 5. oil sunflower plug seedlings; 6. Tekscan 5250-type tactile pressure sensor; 7. collision material; 8. high-speed digital camera.

### 2.3. Ultra-Thin Tactile Pressure Sensor Principle

The Tekscan 5250-type tactile pressure sensor is a matrix-based thin film pressure sensor consisting of two thin polyester sheets featuring patterns of electrically conductive electrodes. The inside surface of one sheet is lined with a number of rows of strip conductors; whereas, the inner surface of the other is lined with a number of columns of strip conductors. The width and spacing of the conductor can be designed to meet different measurement needs. The conductor is coated with a special pressure-sensitive semiconductor material coating. When the two sheets are combined into one, the intersection of a large number of transverse and longitudinal conductors forms an array of pressure-sensing points. When a force is applied to these sensing points, the resistance of the semiconductor changes proportionally to the change in force, thus reflecting the pressure at the sensing point [32,33]. The system's supported I-Scan software enabled collision load information to be displayed in real-time on the PC screen in the form of 2D and 3D contours at a maximum sampling rate of 100 frames per second and enabled the magnitude, temporal characteristics, and location of forces on its surface to be determined. The software also enabled the evaluation of the multicolor presentation of results, resulting pressure and pressure distribution, and statistical data. The sensor of this measurement system has a thickness of 0.1 mm, dimensions of $245.9 \times 245.9$ mm, a spatial resolution of $3.2/cm^2$, and a saturation pressure of 0.179 MPa. A structure diagram of the sensor is shown in Figure 3.

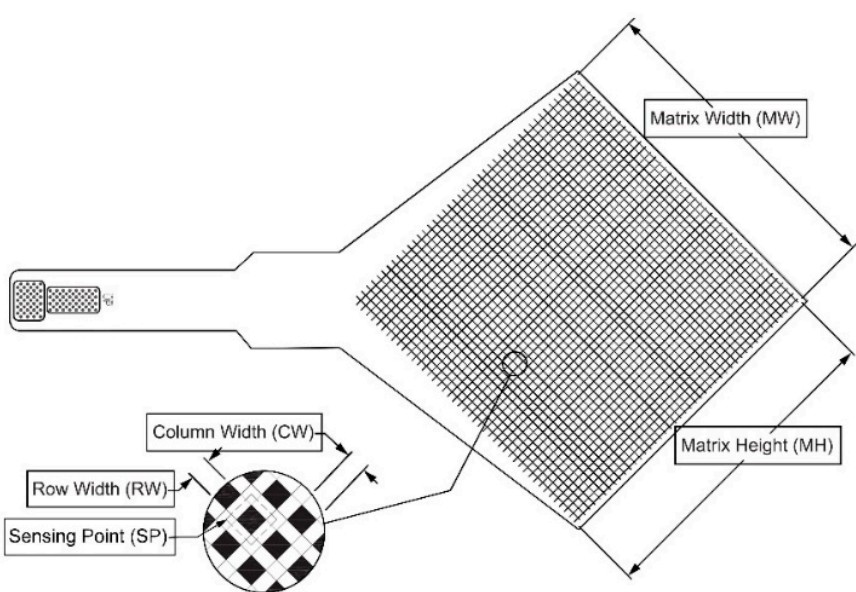

**Figure 3.** Structure diagram of Tekscan 5250-type tactile pressure sensor.

*2.4. Theoretical Analysis of the Seedling Collision Process*

To meet Hertz's contact theory, the following assumptions were made [34–36]: the seedling is regarded as a ball handle with uniform mass; the physical parameters of the plug seedling are consistent [37]; the seedling pot and the collision plate only undergo elastic deformation; the seedling does not undergo relative slip with the collision plate; the friction of contact surface is small, and it is negligible. Figure 4 shows a schematic diagram of the collision between the seedling and the collision plate.

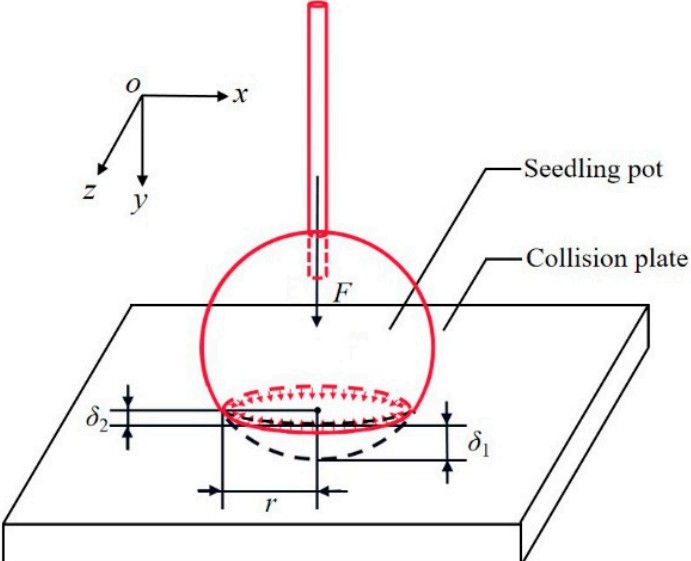

**Figure 4.** Schematic diagram of the collision between the seedling and collision plate: $r$ is the radius of the contact circle, m; $\delta_1$ is the compression of the seedling pot, m; $\delta_2$ is the compression of the collision plate, m; o-xyz is the space coordinate system.

It is known from Hertz's theory that the impact force $F$ is:

$$F = \frac{4}{3} R^{\frac{1}{2}} \delta^{\frac{3}{2}} \left[ \frac{1 - \mu_1^2}{E_1} + \frac{1 - \mu_2^2}{E_2} \right]^{-1} \tag{2}$$

$$\delta = \delta_1 + \delta_2 \tag{3}$$

$$\frac{1}{R} = \frac{1}{R_1} + \frac{1}{R_2} \tag{4}$$

where $F$ is the impact force, N; $R$ is the equivalent radius, m; $R_1$ and $R_2$ are the equivalent radii of the seedling pot and the collision plate, respectively, m; $E_1$ and $E_2$ are the elastic moduli of the seedling pot and the collision plate, respectively, Pa; $\mu_1$ and $\mu_2$ are the Poisson's ratios of the seedling pot and the collision plate, respectively; $\delta$ is the total compression of the seedling pot and collision plate, m.

According to Newton's second theorem [37], the following relation is satisfied:

$$F = -m\frac{d^2\delta}{dt^2} \tag{5}$$

$$\frac{1}{m} = \frac{1}{m_1} + \frac{1}{m_2} \tag{6}$$

As $R_2$ is much larger than $R_1$, $R$ is approximately equal to $R_1$. As $m_2$ is much larger than $m_1$, $m$ is approximately equal to $m_1$. From Equations (2) and (5), the following is derived:

$$-m_1\frac{d^2\delta}{dt^2} = \frac{4}{3}R_1^{\frac{1}{2}}\delta^{\frac{3}{2}}\left[\frac{1-\mu_1^2}{E_1} + \frac{1-\mu_2^2}{E_2}\right]^{-1} \tag{7}$$

Ignoring the effect of air resistance and other factors, the seedling is only affected by gravity during its fall. According to the above assumptions, it is known that only impact force works in the collision process. When the compression of the plug seedling pot and the collision plate reaches the maximum, the impact force reaches the maximum, and the kinetic energy of the plug seedling pot and the collision plate is absorbed by the elastic–plastic deformation, the following equations can be applied:

$$\frac{1}{2}m_1v_0^2 = \int_0^{\delta_{\max}} Fd\delta = \eta F_{\max}\delta_{\max} \tag{8}$$

$$v_0 = \sqrt{2gh} \tag{9}$$

where: $v_0$ is the relative velocity of the seedling and the collision plate at initial collision, m/s; $g$ is the acceleration of gravity, m/s$^{-2}$; $h$ is the drop height, m. $\eta$ is the collision energy absorption coefficient between the plug seedling and collision material.

$\delta_{\max}$ can be obtained from Equations (2) and (8) as:

$$\delta_{\max} = \sqrt[\frac{5}{2}]{\frac{3m_1gh}{4\eta R_1^{\frac{1}{2}}}\left[\frac{1-\mu_1^2}{E_1} + \frac{1-\mu_2^2}{E_2}\right]} \tag{10}$$

Liu et al. [19] believed that the maximum compression is directly related to DSP, that is, the larger the compression, the larger the DSP. Then, the impact peak force (IPF) can be obtained from Equations (2) and (10) as:

$$F_{\max} = \frac{4}{3}R_1^{\frac{1}{2}}\delta_{\max}^{\frac{3}{2}}\left[\frac{1-\mu_1^2}{E_1} + \frac{1-\mu_2^2}{E_2}\right]^{-1} \tag{11}$$

In Equations (10) and (11), $\eta$, $R_1$, $\mu_1$, $\mu_2$, $E_1$ and $E_2$ are all related to the test conditions and contact materials. Therefore, both the DSP and IPF of seedlings are mainly associated with the drop height, collision material, seedling mass, and other factors.

### 2.5. Orthogonal Test Design

The collision material (A), drop height (B), and seedling pot size (C) were taken as the factors, and the drop impact characteristics (IPF) and damage of seedling pot (DSP) were taken as the evaluation indexes of the orthogonal test. The influencing factors and levels are shown in Table 2. Seedling pot size was obtained by selecting different tray sizes of 105, 72, and 50 to obtain three corresponding pot sizes of small, medium, and big. The collision materials were selected from common contact materials used in the transplanting process: steel, ABS plastic and soil block. The drop height was 150 mm, 250 mm, and 350 mm respectively. The orthogonal test was performed using an $L_9(3^3)$ orthogonal table. Each group of drop impact tests was repeated ten times, and the average value was taken as the final value.

**Table 2.** Influencing factors and levels.

| Level | Factors | | |
|---|---|---|---|
| | **Collision Material (A)** | **Drop Height (B)/mm** | **Seedling Pot Size (C)** |
| 1 | Steel | 150 | Small |
| 2 | ABS plastic | 250 | Medium |
| 3 | Soil blocks | 350 | Big |

## 3. Results and Discussion

### 3.1. Analysis of Orthogonal Experiment

The range analysis of the orthogonal experiment is shown in Table 3. The results showed that the significance of the factors that affected the IPF was, from most to least significant, seedling pot size, drop height, and collision material; the factors affecting the DSP were drop height, collision material, and seedling pot size in order of priority. The variance analysis of the orthogonal experiment is shown in Table 4. The results showed that the effects of seedling pot size and drop height on IPF were highly significant, and the effect of collision material on IPF was significant; the effect of drop height on DSP was highly significant, and the effects of collision material and seedling pot size on DSP were significant.

**Table 3.** Experimental arrangement and results.

| Test Number | | Collision Material (A) | Drop Height (B)/mm | Seedling Pot Size (C) | IPF/N | DSP/% |
|---|---|---|---|---|---|---|
| 1 | | 1 | 1 | 1 | 4.60 | 3.18 |
| 2 | | 1 | 2 | 2 | 8.40 | 7.16 |
| 3 | | 1 | 3 | 3 | 10.58 | 10.40 |
| 4 | | 2 | 1 | 2 | 5.12 | 2.45 |
| 5 | | 2 | 2 | 3 | 7.65 | 5.51 |
| 6 | | 2 | 3 | 1 | 5.42 | 5.11 |
| 7 | | 3 | 1 | 3 | 6.50 | 4.09 |
| 8 | | 3 | 2 | 1 | 5.28 | 4.80 |
| 9 | | 3 | 3 | 2 | 8.06 | 8.63 |
| IPF | $k_1$ | 7.86 | 5.41 | 5.10 | | |
| | $k_2$ | 6.06 | 7.11 | 7.19 | | |
| | $k_3$ | 6.61 | 8.02 | 8.24 | | |
| | $R$ | 1.80 | 2.61 | 3.14 | | |
| DSP | $k_1$ | 6.91 | 3.24 | 4.36 | | |
| | $k_2$ | 4.36 | 5.82 | 6.08 | | |
| | $k_3$ | 5.84 | 8.05 | 6.67 | | |
| | $R$ | 2.56 | 4.81 | 2.30 | | |

**Table 4.** Variance analysis of the orthogonal experiment.

| | Factors | Sum of Squares | Degree of Freedom | Mean Squares | *p*-Value |
|---|---|---|---|---|---|
| IPF | A | 5.085 | 2 | 2.542 | 0.019 |
| | B | 10.559 | 2 | 5.279 | 0.009 |
| | C | 15.365 | 2 | 7.683 | 0.006 |
| | *Error* | 0.098 | 2 | 0.049 | |
| DSP | *A* | 9.889 | 2 | 4.944 | 0.029 |
| | *B* | 34.721 | 2 | 17.360 | 0.009 |
| | *C* | 8.596 | 2 | 4.298 | 0.034 |
| | *Error* | 0.299 | 2 | 0.150 | |

Note: $p < 0.01$ (highly significant), $0.01 < p < 0.05$ (significant), $p > 0.05$ (not significant).

*3.2. High-Speed Images Analysis*

Figure 5 shows several key frame images of the seedling collision process; these images represent free fall, collision contact, collision compression, and rebound to the highest position. Before the collision, the seedling fell vertically downward with uniform acceleration (Figure 5a). Due to the difference in the centroid position of the seedling, there is a certain tendency for rotational motion as the seedling falls. At the moment of collision (Figure 5b), the seedling pot came into contact with the collision material, and the seedling pot gradually experienced compression deformation. When the compression deformation reached its maximum (Figure 5c), the relative motion velocity of the seedlings became zero, and the kinetic energy of the seedlings was completely converted into the elastic–plastic strain energy of the seedling pot. By comparing Figure 5c with 5d, it can be concluded that when the seedling pot underwent maximum compression deformation, the stored elastic–plastic strain energy began to be gradually released. Some of the energy was converted into elastic potential energy, which resulted in the rebound of the seedlings. The other part of the energy was absorbed by the compression deformation of the seedling pot and collision plate, eventually leading to energy dissipation. Damage to the seedling pot caused by the collision compression is shown in Figure 5d. The above analysis shows that damage to the seedling pot has a significant relationship with the compression deformation of the pot and the buffering performance of the collision material.

*3.3. Analysis of Single-Factor Experiments*

From the results of the orthogonal test, it can be concluded that collision material, drop height, and seedling pot size were the significant factors that affected the IPF and DSP. In order to clarify the significant factors that affected the impact characteristics of the seedlings, the single-factor test was conducted for the above-mentioned significant factors. The change rule of contact pressure distribution and contours during the collision was analyzed. The IPF and DSP of each test were determined, respectively. Each single-factor experiment was repeated ten times, and the average value was taken as the final result.

3.3.1. Collision Material and Drop Height

Steel, ABS plastic, and soil blocks were selected as the collision materials. The drop heights of 50 mm, 150 mm, 250 mm, and 350 mm were selected. The seedling pot size was small. The results of the single-factor test are shown in Figure 6. With the increase in drop height, both IPF and DSP gradually increased. The possible reason was: the higher the drop height, the greater the impact force of the seedling on the collision materials, and the greater the reaction force of the seedling, resulting in some squeezing pressure, friction and gravity to increase the elastic plastic deformation of the seedling pot and aggravate pot damage. Therefore, to reduce the collision damage of the seedling pot during the transplanting operation, the height difference between the seedling and collision materials should be considered [35]. Among the three collision materials, both IPF and DSP between the seedling and steel, soil block, and ABS plastic decreased sequentially. The buffering performance of steel and ABS plastic materials is poor. Soil block material has a buffering

effect and a certain hardness. Its strain energy stored in the compression stage was easily transmitted to the surroundings in the form of stress waves, which led to reduced seedling pot damage.

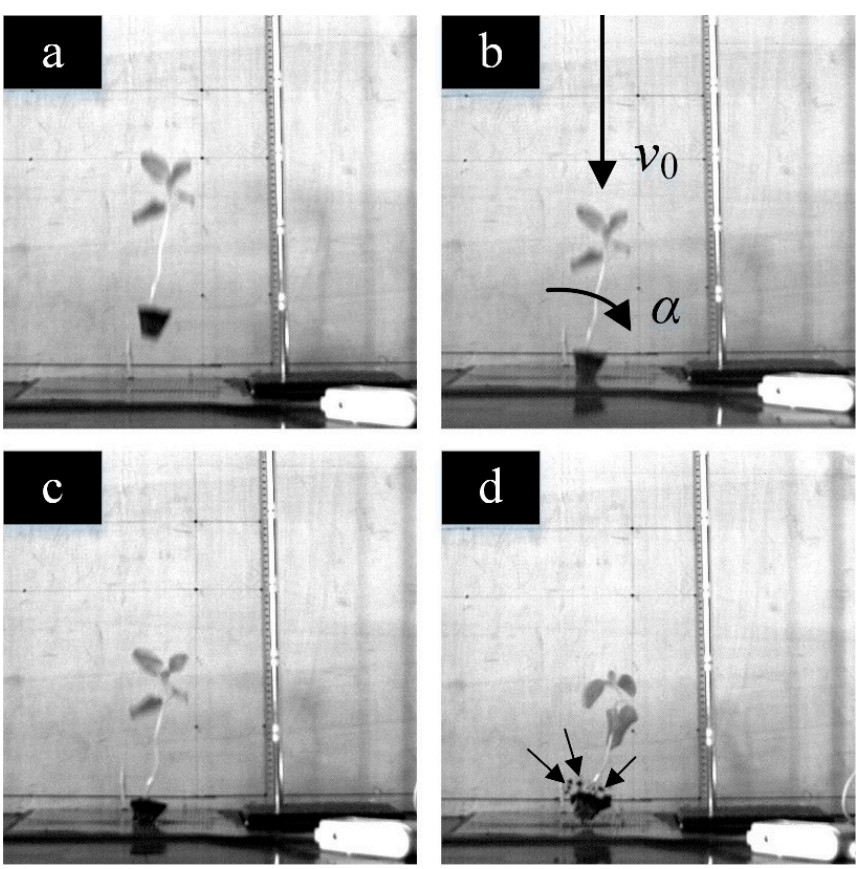

**Figure 5.** High-speed images of seedling collision: (**a**–**d**) represent the free drop, collision contact, collision compression, and rebound to the highest position of plug seedlings, respectively; α represents the rotation angle, defined as the angle between the stem and $v_0$ velocity direction, (°). The collision material is steel plate, the drop height is 250 mm, and the seedling pot size is medium.

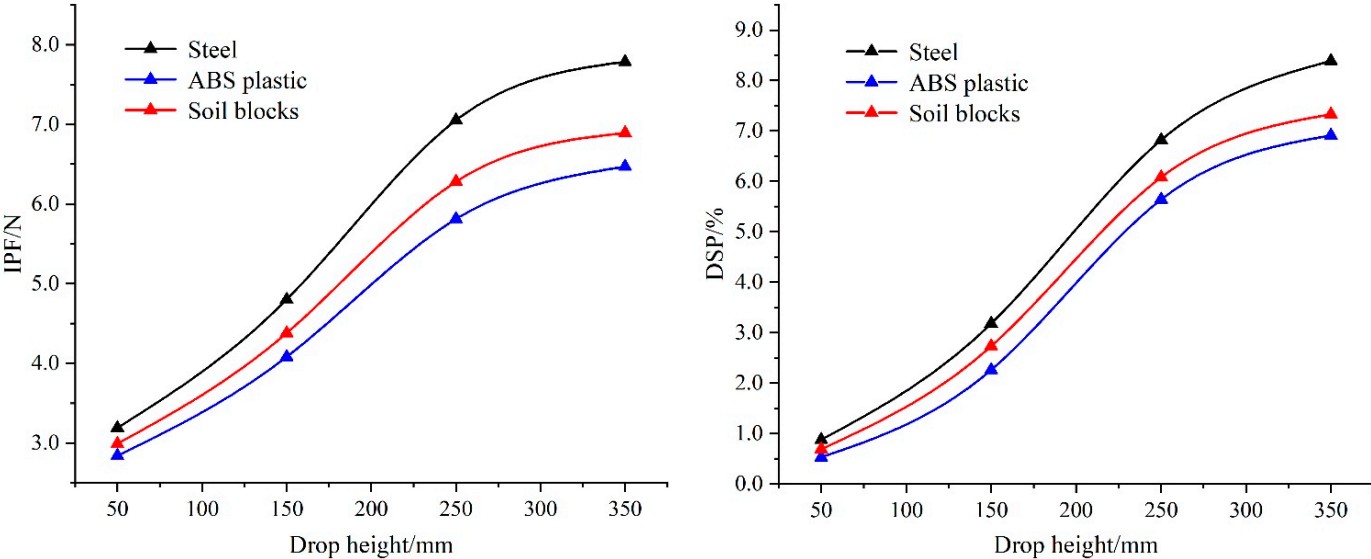

**Figure 6.** Effect of collision materials and drop heights on IPF and DSP.

Pressure distribution contours at different heights for the three collision materials are shown in Figure 7. By observing the 2D contours, it can be seen that high pressure is distributed at the center of the pressured area, which is surrounded by a regularly extended contour. Low pressure is distributed at the edge of the pressured area. From drop heights of 50, 150, 250, 350 mm, the pressured areas were 182, 218, 273, and 326 mm² for seedling impacts against steel material, and the pressured areas were 188, 243, 281, and 352 mm² for seedling impacts against soil blocks material, and the pressured areas were 177, 212, 257, and 298 mm² for seedling impacts against ABS plastic material. Obviously, with the continuous increase of drop height, the pressured area between the seedling pot and the different collision material gradually becomes larger; among the collision materials, the pressured area between the seedling pot and the soil block was the largest. Materials with poor buffering performance such as steel and ABS plastic materials have extremely limited deformation ability so the contact pressure distribution area of plug seedlings is mainly determined by the deformation of the seedling pot itself. Therefore, the plug seedling with a large contact pressure distribution area will have relatively serious seedling pot damage. The result was the same as that in Section 3.3.1. In comparison, the contact pressure distribution area between the plug seedling and the soil block material is obviously higher, but the damage of the seedling pot is not the most serious. The reason is that when the plug seedling collides with the soil block, the soil block can expand the contact area with the seedling pot through its own deformation, thereby absorbing the impact energy to reduce the damage degree of the seedling pot. At the same time, with the continuous increase of drop height, the high-pressure distributed area showed a gradually increasing trend indicating that with the increase of falling height, the compression deformation of the seedling pot gradually increased, and thus the seedling pot was more likely to be damaged.

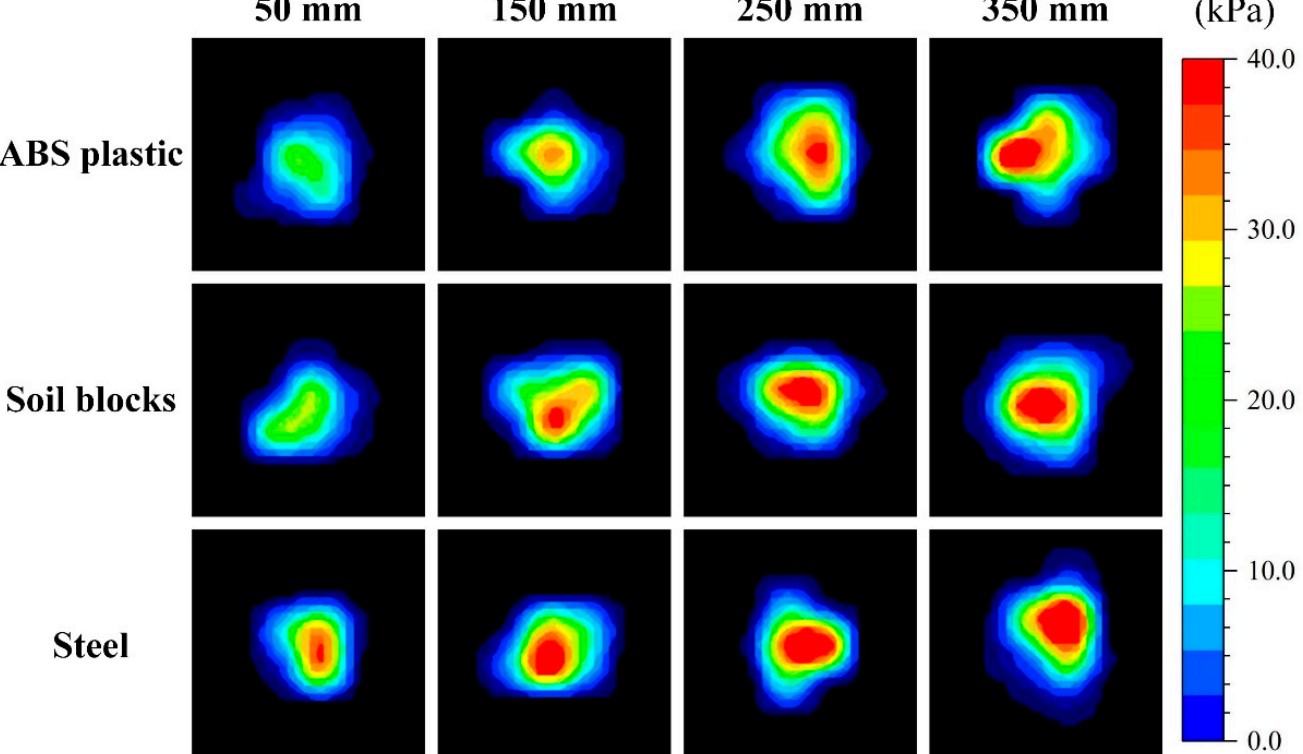

**Figure 7.** Pressure distribution contours at different drop heights for three collision materials.

Figure 8 shows the pressured area distribution for impacts against the three collision materials. In the case of the seedling dropping on the three impact surfaces, when the drop height of the plug seedling is low, the pressured area with a contact pressure of 0~10 kPa accounted for the highest proportion. As the height of the fall increased, although 0~10 kPa

contact pressure still maintains a large pressured area, the proportion began to significantly reduce. The changing pattern of the steel material is the most obvious. This shows that with the increase in the drop height, the contact pressure (10~40 kPa) accounts for the main pressured area, which is the main reason for the collision damage of the seedling pot [38].

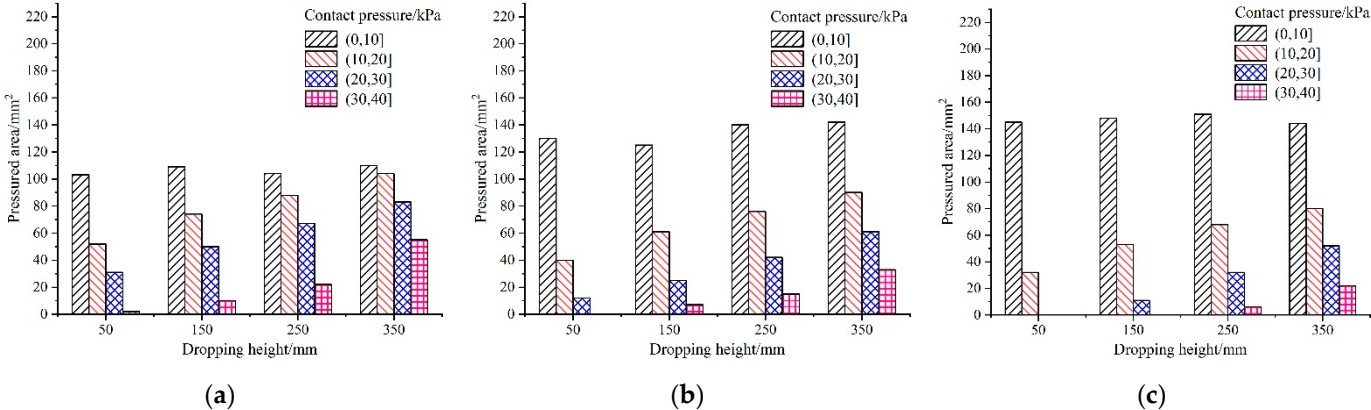

**Figure 8.** Pressured area distribution at different drop heights for three collision materials: (**a**) steel, (**b**) soil blocks, and (**c**) ABS plastic.

### 3.3.2. Seedling Pot Size and Drop Height

The collision material was steel. The drop heights of 50 mm, 150 mm, 250 mm, and 350 mm were selected. The small, medium and big sizes of the seedling pot were selected. The results of the single-factor test are shown in Figure 9. The DSP of the seedling pot of big, medium, and small sizes decreased sequentially. The main reason for this was that when the seedling age was 25 days, the roots of seedlings with a big pot size were not sufficiently twined with the substrate, which reduced the damage resistance of the seedling pot. For seedlings with medium- and small-sized pots, the seedling roots were able to more fully twine and contact with the substrate at the same seedling age, thus resulting in a higher damage resistance of the seedling pot. Among the three seedling pot sizes, the IPF of the big pot was the largest, and that of the small seedling pot was the smallest. The greater the seedling mass was, the larger the inertial force generated by collision and the plastic deformation of the seedling pot would be, which resulted in the impact force of the seedling on the collision materials increasing during the collision [39].

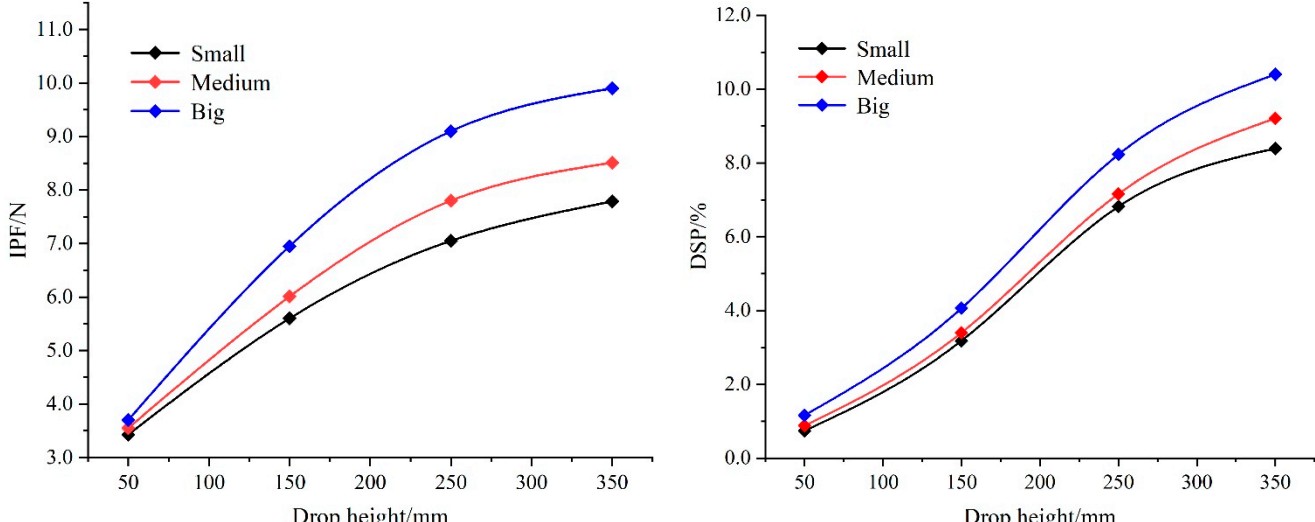

**Figure 9.** Effect of seedling pot size and drop heights on IPF and DSP.

Pressure distribution contours at different heights for the seedlings of three pot sizes are shown in Figure 10. From drop heights of 50, 150, 250, and 350 mm, the pressured areas were 182, 218, 273, and 326 mm$^2$ for seedlings with a small pot size, and the pressured areas were 199, 286, 352, and 406 mm$^2$ for seedlings with a medium pot size, and the pressured areas were 275, 368, 373, and 431 mm$^2$ for seedlings with a big pot size. It can be seen that the pressured area of seedlings of different pot sizes was big, medium, and small in descending order when the drop height was constant. A possible reason could be the larger the size of the seedling pot, the higher the number of pores and the increased deformation capacity of the seedling pot. Therefore, in the collision process, the seedling pot will rely on its own deformation to increase the contact area with the collision materials.

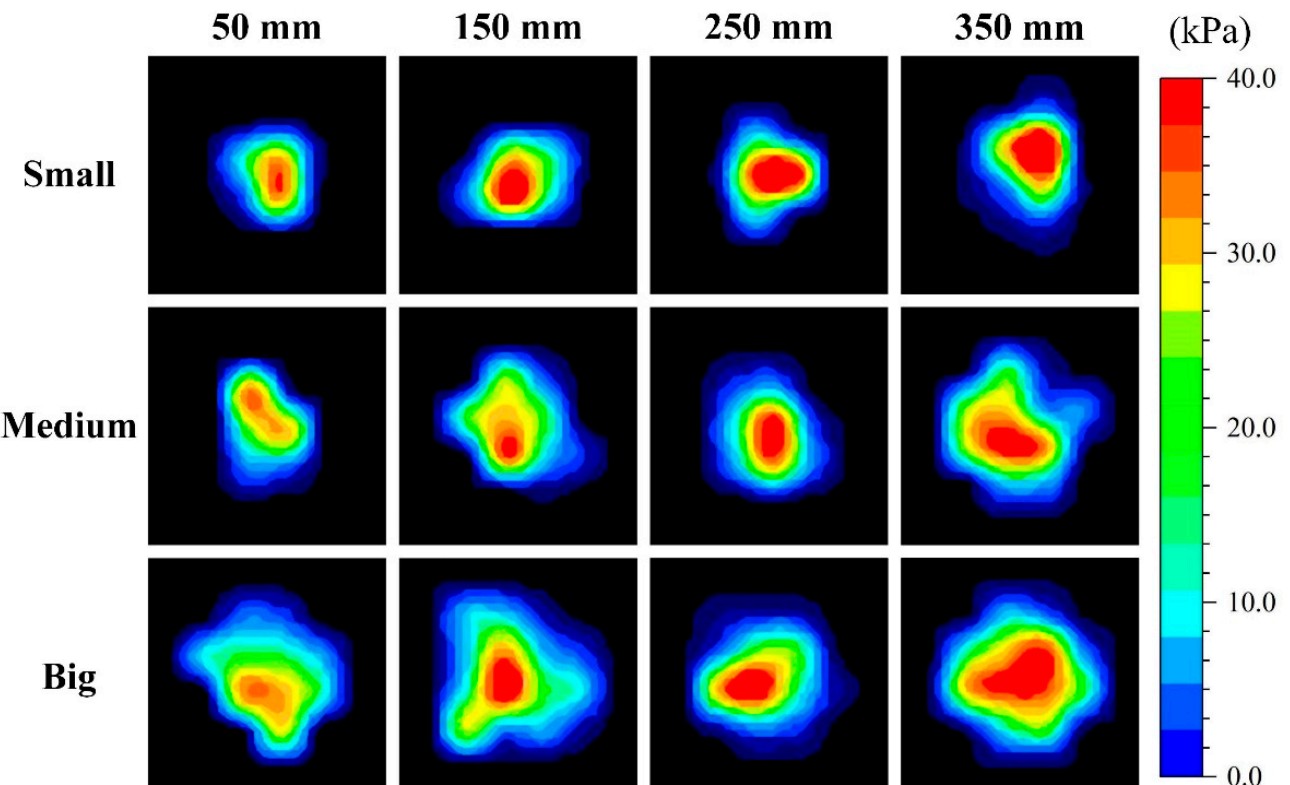

**Figure 10.** Pressure distribution contours at different drop heights for three seedling pot sizes.

### 3.3.3. Establishment of Damage Prediction Model

Figure 11 shows the IPF-DSP relationship fitted by linear regression equations in the case of seedlings dropping on the three impact surfaces and the seedlings of different pot sizes dropping on steel surfaces. Linear regression models were obtained, and the coefficients of determination ($R^2$) for IPF-DSP were 0.98 and 0.94, respectively. This shows that the feasibility of assessing and predicting seedling pot damage by means of the flexible film-network tactile pressure sensor are demonstrated. Meanwhile, according to Equation (11), when the plug seedling, drop height, and collision material are determined, the values of $\eta$, $R_1$, $\mu_1$, $\mu_2$, $E_1$ and $E_2$ are also determined. Therefore, DSP has a linear relationship with IPF. This is consistent with the results of the regression equation.

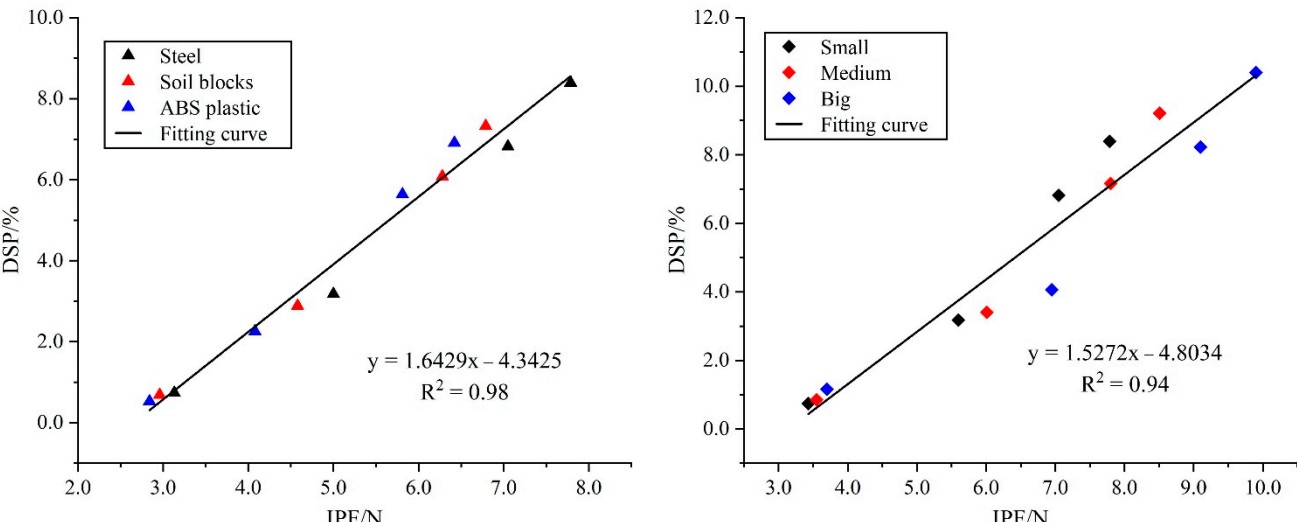

**Figure 11.** IPF and DSP relationship for different materials and seedling pot sizes.

## 4. Conclusions

In this study, the contact mechanics models of the plug seedling were established for seedling dropping based on the Hertz theory. The feasibility of assessing and predicting seedling pot damage by means of the Tekscan pressure distribution testing technique was demonstrated, and the relationship between contact pressure distribution and seedling pot damage was determined, which provides a new method and theoretical basis for studying the mechanical properties of collision contact during transplanting, and which also provides an essential reference for other agricultural material impact studies. The main conclusions are as follows:

(1) Based on the orthogonal tests, the influence laws of various factors on IPF and DSP were obtained. The results showed that the order of influencing factors for IPF was found to be: seedling pot size > drop height > collision material. The order of influencing factors for DSP was found to be: drop height > collision material > seedling pot size;

(2) The Tekscan pressure distribution measurement system measured the change law of contact pressure distribution under significant influencing factors. The relationship between contact pressure distribution and seedling pot damage was determined. The result showed that with the increase of drop height, the pressured area between the seedling pot and the collision material increased gradually, and the high-pressure area showed a gradually increasing trend. In descending order, the pressured area of seedlings of different pot sizes was big, medium, and small. The pressured area of different materials was soil blocks, steel, and ABS plastic in descending order;

(3) Linear regression models between IPF and DSP under different factors were established based on the pressured data collected by the Tekscan pressure distribution testing system, and the determination coefficients ($R^2$) were 0.98 and 0.94, respectively.

**Author Contributions:** Conceptualization, X.L.; methodology, H.B. and F.Z.; software, J.C.; formal analysis, Y.Z.; formal analysis and writing—original draft preparation, writing review and editing, H.B. All authors have read and agreed to the published version of the manuscript.

**Funding:** This research was funded by the National Natural Science Foundation of China (No. 32160423) and Natural Science Foundation of the Inner Mongolia Autonomous Region of China (2021MS05048).

**Data Availability Statement:** The data presented in this study are available on demand from the first author at (bhb81571@163.com).

**Acknowledgments:** The authors gratefully acknowledge the financial support provided by the National Natural Science Foundation of China (No. 32160423). We also appreciate the work of the editors and reviewers of this paper.

**Conflicts of Interest:** The authors declare no conflict of interest.

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
