# Peer review of "Study on the Impact Damage Characteristics of Transplanting Seedlings Based on Pressure Distribution Measurement System"

_horticulturae, doi:10.3390/horticulturae8111080_

Round 1

Reviewer 1 Report

In this manuscript, the kinetics equations of the seedling collision is provided based on Hertz's contact theory, and the kinematics characteristics, elastoplastic deformation, and collision damage during seedlings collision is analyzed using high-speed photography.  The paper is well organized and includes new contributions. However, a number of issues/errors in the manuscript are expected to be solved.

- More detailed explanation of the work and a brief of the results should be considered at the end of the introduction.

In the introduction, the damage characteristics and mechanical properties of  plug seedlings are introduced by using FEM and DEM. However, the ‎authors should appropriately extend this section by discussing more relevant works focusing on ‎the effect of contact conditions on the damage mechanisms and the strength of materials. For example, it is suggested to read and discuss ‎the following relevant works:‎

DEM Modeling of Crushable Grain Material under Different Loading Conditions, Periodica Polytechnica Civil Engineering, 65(3), pp. 935–945, 2021. https://doi.org/10.3311/PPci.17948

- It will be more appropriate if you add a paragraph at the end of the introduction section illustrating the layout of the paper.

-On page 5 line 157, E parameter is introduced, however, in the equation is not considered.

-On page 6 line 171, please correct the error.

-On page 7, Figure 5 (A-D) in the text should be capital according to the Figure 5.

- Please explicitly declare the assumptions and limitations of the models.

-In the conclusion please clearly mention the practical outcome of the study. Please clarify what the employed technique offers to the engineering field compared to other methods available in the literature.

Reviewer 2 Report

It is a very interesting experiment and accurately describes the degree of nursery damage in terms of physical parameters.

However, the conclusions drawn in this paper only summarize the results of various experiments and leave out some considerations.

For example, only one type of soil was used in this study, and it is unclear how water content and hardness relate to the results. If a theoretical model was created, shouldn't the characteristics of the materials used also be considered?

It is also unclear what effect this nursery damage has on seedling growth, and it is not clear whether the results of this experiment can be used.

Please add the above points and revise the paper to make it more useful.

(Minor correction)

Figure 2 is difficult to see, so please put a larger photo or create an image.

In the caption of Figure 5, (A), (B), (C), and (D) should be in lower case.

Reviewer 3 Report

It is interesting attempt. But the submitted manuscript is far from original paper.

It is quite natural that the seedling is damaged by the dropping. The result on the effects following the collision material, drop height, and seedling pot size on impact peak force are also natural as a physical principle. Hertz contact theory is also described in the document. But the relationship between the delived model and the result of the experiment is not disuccsed. It is not revealed how each faftor affet the impact force and damage of seedling pot. The resulted data might be valuable. But threre is no scientific merit. Beseide I could not find out  how valuable to imporove the transplanting opereaion of seedlings by these results.
